# The Gram-Positive Bacterium *Leuconostoc pseudomesenteroides* Shows Insecticidal Activity against Drosophilid and Aphid Pests

**DOI:** 10.3390/insects11080471

**Published:** 2020-07-25

**Authors:** Nils Hiebert, Tobias Kessel, Marisa Skaljac, Marius Spohn, Andreas Vilcinskas, Kwang-Zin Lee

**Affiliations:** 1Fraunhofer Institute for Molecular Biology and Applied Ecology, Ohlebergsweg 12, D-35394 Giessen, Germany; nils.hiebert@ime.fraunhofer.de (N.H.); marisa.skaljac82@gmail.com (M.S.); marius.spohn@ime.fraunhofer.de (M.S.); andreas.vilcinskas@agrar.uni-giessen.de (A.V.); 2Institute for Insect Biotechnology, Justus-Liebig University of Giessen, Heinrich-Buff-Ring 26, D-35392 Giessen, Germany; tobias.kessel@agrar.uni-giessen.de

**Keywords:** invasive species, insect pests, biological control, *Leuconostoc* spp.

## Abstract

Insect pests reduce global crop yields by up to 20%, but the most effective control measures are currently based on environmentally hazardous chemical pesticides. An alternative, ecologically beneficial pest-management strategy involves the use of microbial pathogens (or active compounds and extracts derived from them) that naturally target selected insect pests. A novel strain of the bacterium *Leuconostoc pseudomesenteroides* showed promising activity in our preliminary tests. Here, we investigated its effects in more detail, focusing on drosophilid and aphid pests by testing the survival of two species representing the family Drosophilidae (*Drosophila suzukii* and *D. melanogaster*) and one representing the family Aphididae (*Acyrthosiphon pisum*). We used oral and septic infection models to administer living bacteria or cell-free extracts to adult flies and aphid nymphs. We found that infection with living bacteria significantly reduced the survival of our insect models, whereas the administration of cell-free extracts had a significant effect only in aphids. These results confirm that *L. pseudomesenteroides* has potential as a new biocontrol agent for sustainable pest management.

## 1. Introduction

Insect pests damage plants by direct feeding and by vectoring pathogens [1,2,3,4]. In this manner, agricultural pest insects reduce global crop yields by up to 20% per year [5,6]. Agrochemicals are widely used for the control of insects to protect food/feed crops, and although some chemical insecticides are not hazardous when applied correctly, many harm beneficial insects (such as pollinators and natural predators) and their use is now banned by statute [7]. The widespread use of pesticides has also increased selection pressure, leading to the emergence and spread of resistant populations. We therefore need new pest control methods that cause less environmental harm and are compatible with sustainable agriculture [8,9]. One alternative strategy that addresses these challenges is the use of biological control agents, including natural pathogens and their specific bioactive products. 

Biological control agents have been used in agriculture for many decades, beginning with entomopathogenic microbes such as *Bacillus thuringiensis* and *Beauveria bassiana* [10,11]. These widely used biopesticides [12,13] control a range of target insects [14] and the market for them is forecast to grow by 10% annually [15]. More recently, some strains of *Burkholderia* spp. [16,17,18,19,20,21] have shown promising broad-spectrum insecticidal properties and the natural compound spinosad from *Saccharopolyspora spinosa* [22] was efficacious against a variety of insect pests [23,24,25]. The role of insect-pathogenic bacteria as a component of integrated pest management has been reviewed [26]. We have recently demonstrated the efficacy of biocontrol agents for the management of certain insect pests [27,28,29,30] and currently focus on the discovery and analysis of novel entomopathogenic microbes that show activity against fruit pests such as *Drosophila suzukii* and *D. melanogaster*, as well as aphid pests such as *Acyrthosiphon pisum*. 

The spotted-wing drosophila (*D. suzukii*) is one of the most damaging pests of soft-skinned fruits. It originates from Asia [31] but has spread to America and Europe as an invasive species that is notoriously difficult to control [32,33,34,35,36]. This reflects its use of a saw-like serrated ovipositor to pierce the skin of ripening fruits and lay eggs in the flesh immediately before harvesting [37,38]. In contrast, the model fruit fly *D. melanogaster* is attracted to rotting fruits and thus has a much lower impact on agriculture [39,40], although there is evidence that it vectors the microbes that cause sour rot disease in grapes [41,42]. *D. melanogaster* is better known as a laboratory model organism [43] and is often used for the analysis of host–microbe interactions [44,45,46]. We included these two drosophilids in our study to investigate differences in the virulence of entomopathogens toward different pest species in the same genus. Finally, we included *A. pisum* as a model aphid [30,47]. Aphids penetrate plant tissues and feed on the phloem [48], reducing host fitness and yields by depleting nutrients and transmitting pathogens [1,2,3,4]. All three species have a short generation cycle and a high reproductive rate to facilitate multigenerational experiments. Both *D. suzukii* and *A. pisum* are also economically important pests with a wide host range and pesticide-resistant populations, making them key targets for novel biocontrol strategies [2,3,4,41,42,49,50,51,52,53].

As part of our ongoing effort to isolate new entomopathogenic bacteria, we identified a *Leuconostoc pseudomesenteroides* strain that strongly inhibited the growth of drosophilids in our initial screening assay [54]. *Leuconostoc* spp. are important lactic acid bacteria widely used as starter cultures for the fermentation of vegetables and dairy products, with potential applications as functional food ingredients [55,56,57]. Beneficial and food grade lactic acid bacteria with qualified presumption of safety status could therefore provide a valuable source of alternative biocontrol agents [58]. In this study, we investigated the ability of *L. pseudomesenteroides* cells and cell-free extracts to inhibit the survival of our three target insects via the oral and septic infection routes. Although lactic acid bacteria are known to interfere with the growth of phytopathogenic bacteria and fungi [59,60], and also entomopathogenic bacteria [61], little is known about their potential as biocontrol agents against insect pests [62].

## 2. Materials and Methods 

### 2.1. Maintenance of Drosophila Species

We maintained *D. suzukii* specimens (originally from Ontario, Canada) and the *D. melanogaster* strain *white* (*w*) A 5001 in climate chambers under constant conditions (12-h photoperiod, 26 °C and 60% relative humidity). Larvae and adult flies were fed on a soybean/cornmeal medium comprising 10.8% (*w*/*v*) soybean and cornmeal mix, 0.8% (*w*/*v*) agar, 8% (*w*/*v*) malt, 2.2% molasses, 1% (*w*/*v*) nipagin and 0.625% (*w*/*v*) propionic acid. All experiments were carried out using post-eclosion female flies aged 3–10 days.

### 2.2. Maintenance of Aphids

We maintained *A. pisum* clone LL01 in a climate chamber with a 16-h photoperiod (24 °C day, 18 °C night) and provided the host plant *Vicia faba* var. *minor* (2–3 weeks old) as a food source [19]. Nymphs were synchronized by transferring breeding aphids to a *V. faba* leaf in a Petri dish containing 1.5% agar and collecting newborn nymphs after 24 h [30]. They were reared for another 4 days so that 5-day-old age-synchronized nymphs could be used for all experiments.

### 2.3. Cultivation of L. pseudomesenteroides and Preparation of Extracts

The *L. pseudomesenteroides* strain was isolated from moribund *D. suzukii* larvae as previously described [54]. Pre-cultures of *L. pseudomesenteroides* were prepared in 100-mL Erlenmeyer flasks containing 30 mL MRS medium (10 g/L casein peptone tryptic digest, 10 g/L meat extract, 5 g/L yeast extract, 20 g/L glucose, 1 g/L Tween-80, 2 g/L K_2_HPO_4_, 5 g/L sodium acetate, 2 g/L ammonium citrate, 0.2 g/L MgSO_4_ × 7H_2_O, 0.05 g/L MnSO_4_ × H_2_O). The cultures were inoculated from a glycerol stock and incubated overnight at 28 °C, shaking at 180 rpm.

For the preparation of live bacterial cultures for the oral and septic assays, 0.5 mL of the pre-culture were transferred to a 300-mL Erlenmeyer flask containing 50 mL MRS medium and incubated overnight as above before centrifugation and resuspension in sucrose for fly feeding (Section 2.4), advanced AP3 medium [63] for aphid feeding (Section 2.6) or phosphate-buffered saline (PBS) for injection in both species (Section 2.5 and Section 2.7).

For the preparation of cell-free extracts, 0.5-mL aliquots of the pre-culture were transferred to 300-mL Erlenmeyer flasks containing 50 mL MRS medium as above, and incubated for 24, 48 or 96 h. In each case, the broth was lyophilized and extracted with methanol. The lyophilized culture was incubated for 2 h with methanol while shaking. Afterwards, cell debris was separated by centrifugation and extract supernatants were transferred to a fresh vessel for drying under reduced pressure. To confirm the absence of cells, an aliquot of the extract reconstituted in 100% DMSO was incubated on MRS agar plates. No colonies grew after overnight incubation at 28 °C. Negative control extracts were prepared from sterile MRS broth. For the drosophilid feeding assay, dried extracts were reconstituted to a 100× concentrated stock solution in DMSO based on the initial culture volume. The stock solution was then diluted to a 1× working solution in 50 mM sucrose. For the aphid feeding assay, dried extracts were reconstituted to a 1× working solution in advanced AP3 medium. For all injection experiments, dried extracts were reconstituted to a 1× working solution in PBS.

### 2.4. Feeding of Drosophila Species (Oral Infection Route)

For the live bacteria feeding assays, overnight bacterial cultures were centrifuged at 1700× *g* for 10 min at 4 °C and the cell pellet was resuspended in 2 mL 50 mM sucrose to an optical density (OD_600_) of 1 or 0.1 as required. *L. pseudomesenteroides* cell counts revealed that an OD_600_ of 1 corresponded to 4.9 × 10^8^ CFU/mL. The suspensions were transferred to vials (diameter = 2.5 cm) lined with three layers of paper towels. Female flies of each species (20 per experiment) were transferred to the vials and maintained under the conditions described above. Negative control vials were also prepared in which the flies were presented with 50 mM sterile sucrose. Surviving flies were counted daily and 200 µL 100 mM sucrose were added after counting to replace nutrients.

For the cell-free extract feeding assays, 20 female flies were starved for 5 ± 1 h before being transferred to vials containing Parafilm loaded with 12 droplets (3 µL each) of the cell-free *L. pseudomesenteroides* extracts or negative control extracts prepared from the sterile MRS medium (1× working solution in 50 mM sucrose). After allowing the flies to feed for 2 h, they were transferred to fresh vials lined with paper towels soaked with 2 mL 50 mM sucrose. Surviving flies were counted daily and 200 µL 100 mM sucrose was added after counting to replace nutrients.

All survival assays were performed independently at least three times (biological replicates) using three cohorts consisting of 20 female flies each (technical replicates). 

### 2.5. Injection of Drosophila Species (Septic Infection Route)

Overnight bacterial cultures or extracts corresponding to 4.9 × 10^8^ CFU/mL (24, 48 and 96 h) reconstituted in PBS were transferred to 1.5-mL Eppendorf tubes, and 4.6 nL were introduced into female flies by intrathoracic injection using a Nanoject II device Drummond Scientific, Broomall, PA, USA). Mock injections were carried out as negative controls. For the live cell injections, the negative controls were mock injections with the same volume of PBS. For the extract injections, the negative controls were mock injections with the same volume of extract prepared from sterile MRS culture medium and reconstituted in PBS. After injection, flies were transferred to vials lined with paper towels soaked in 2 mL 50 mM sucrose and were maintained and counted as described above.

### 2.6. Feeding of Aphids (Oral Infection Route)

For the live bacteria feeding assays, overnight bacterial cultures were centrifuged at 1700× *g* for 10 min at 4 °C and the cell pellet was resuspended in 2 mL advanced AP3 medium to an OD_600_ of 1 or 0.1 (4.9 × 10^8^ and 4.9 × 10^7^ CFU/mL, respectively) as required. Negative controls were prepared in which the aphids were presented with the sterile advanced AP3 medium alone. For the cell-free extract feeding assays, 1× working solutions in advanced AP3 medium were transferred to the base of small vials between two layers of Parafilm. Five age-synchronized aphids (5 days old) were introduced into each vial. The small vials were placed upside down in 24-well plates. Aphids were able to pierce the Parafilm and suck up the bacterial suspensions or extracts. Negative controls were prepared using the extract derived from the sterile MRS medium. 

All survival assays were performed independently at least three times (biological replicates) with 60 aphids per assay. Survival was scored daily over the 3 days of exposure. 

### 2.7. Injection of Aphids (Septic Infection Route)

Overnight cultures and extracts of *L. pseudomesenteroides* were reconstituted in PBS as described above. Aphids were held in place by applying a vacuum and were injected ventrally between the hind legs [19]. We injected 25 nL of the bacterial suspension or the same quantity of 1× cell-free extract using an M3301 micromanipulator (World Precision Instruments) with glass capillaries. For the live cell injections, the negative controls were mock injections with the same volume of PBS. For the extract injections, the negative controls were mock injections with the same volume of extract prepared from sterile MRS culture medium and reconstituted in PBS. We injected 20 aphids per treatment in three biological replicates. Injected aphids were placed individually for 10 days in Petri dishes containing *V. faba* leaves on a 1.5% agar substrate [19] and survival was checked daily. Petri dishes with leaves were replaced after 5 days to ensure the aphids were maintained under optimal conditions.

### 2.8. Data Analysis

The identity of *L. pseudomesenteroides* was confirmed by 16*S* rDNA sequencing after PCR amplification using forward primer 5′-AGA GTT TGA TCM TGG CTC AG-3′ and reverse primer 5′-CGG TTA CCT TGT TAC GAC TT-3′ and REDTaq ReadyMix (Sigma-Aldrich, Hamburg, Germany) according to the manufacturer’s instructions. The PCR product was sequenced by Eurofins Genomics and searched against the online databases on EzBiocloud (http://ezbiocloud.net/). The 16S rRNA sequence was submitted to GenBank (accession number MT24349). Survival curves were plotted and evaluated by Kaplan–Meier logrank analysis using GraphPad Prism (*p* ≤ 0.0009 ***; *p* ≤ 0.009 **; *p* ≤ 0.09 *).

## 3. Results

### 3.1. Oral Infection of Drosophilids with L. pseudomesenteroides 

To determine the insecticidal activity of *L. pseudomesenteroides* in *D. melanogaster* and *D. suzukii*, we fed adult female flies with bacteria reconstituted in 50 mM sucrose at an OD_600_ of 1 or 0.1. We found that *D. melanogaster* females infected at OD_600_ = 1 died significantly faster than the sucrose controls (*p* < 0.0001) in all three biological replicates (Figure 1A–C) with an average LT_50_ of 5 days. Those infected at OD_600_ = 0.1 succumbed faster in two replicates (Figure 1A,B) with an LT_50_ of 12 days but there was no significant difference compared to the sucrose control in the third replicate (Figure 1C). In contrast, infected *D. suzukii* females died significantly faster than the sucrose controls (*p* < 0.0001) at both OD_600_ values, confirming the efficacy of *L. pseudomesenteroides* against this species (Figure 1D–F). 

### 3.2. Septic Infection of Drosophilids with L. pseudomesenteroides

Next, we tested the effect of injecting *L. pseudomesenteroides* reconstituted in PBS (OD_600_ = 1 or 0.1) into the thorax of female flies. In *D. melanogaster*, the first biological replicate indicated a significant difference in survival (*p* < 0.0001) between the flies injected with bacteria and the mock-injection PBS controls, regardless of the bacterial concentration (Figure 2A). However, the second replicate revealed no significant effects (Figure 2B) and the third replicate showed a minor (albeit significant) effect at OD_600_ = 0.1 but no significant effect at OD_600_ = 1 (Figure 2C). Altogether, no clear effect was detected for *D. melanogaster*. In contrast, *D. suzukii* showed a significant (*p* < 0.0001–0.0003) response to the injection of *L. pseudomesenteroides* at both concentrations in all three replicates (Figure 2D–F). Interestingly, the injection of bacteria significantly prolonged survival in the first replicate (Figure 2D) but inhibited survival in the remaining replicates, resulting in an LT_50_ range of 11–12 days (Figure 2E,F). There was no significant difference in survival between *D. suzukii* cohorts injected at OD_600_ 1 and 0.1.

### 3.3. Oral Infection of A. pisum with L. pseudomesenteroides

Next, we investigated the effect of *L. pseudomesenteroides* against *A. pisum* in oral infection assays. Aphids were fed on bacterial suspensions reconstituted in advanced AP3 medium at OD_600_ = 1 or 0.1, or on the sterile AP3 medium as a control. Survival was determined after 3 days (Figure 3A). Infection caused a significant 25–30% decline in survival at both OD_600_ = 1 (*p* = 0.0005) and OD_600_ = 0.1 (*p* < 0.0001) and the overall effect was similar at both doses.

### 3.4. Septic Infection of A. pisum with L. pseudomesenteroides

Finally, we investigated the effect of *L. pseudomesenteroides* against *A. pisum* by septic infection, tracking the survival of the aphids until 10 days post injection (Figure 3B). The injected aphids showed a rapid decline in survival compared to the PBS mock-injection controls, resulting in 100% mortality within 3 days (*p* < 0.0001). The lethal effect of the injection was similar at both OD_600_ values.

### 3.5. Oral Administration of L. pseudomesenteroides Extracts to Drosophilids

To determine whether the effect of *L. pseudomesenteroides* requires live bacteria or is conferred by soluble toxins, methanol extracts of the bacterial cultures after 24, 48 and 96 h were dried, reconstituted as a stock in DMSO and diluted to a 1× working solution in 50 mM sucrose. The extracts were presented to starved flies of both species. *D. melanogaster* adult females fed on the 24-h extract died significantly faster (*p* < 0.0001) than MRS extract controls (LT_50_ = 15–16 days), whereas the 48-h extract had no significant effect and the 96-h extract showed a weak but still significant (*p* = 0.0033) effect (Figure 4A). The bacterial extracts had no significant negative effect on *D. suzukii* regardless of the preparation time. Only the 24-h extract showed a statistically significant effect (*p* = 0.0022) but this prolonged survival compared to the control (Figure 4B). These data suggest the orally administered *L. pseudomesenteroides* extracts have a negligible effect in drosophilids.

### 3.6. Septic Administration of L. pseudomesenteroides Extracts to Drosophilids

We also injected flies with the *L. pseudomesenteroides* extracts reconstituted to a 1× working solution in PBS. In *D. melanogaster*, we observed a significant effect (*p* = 0.0047) following the injection of the 24-h extract but not the others (Figure 5A). In *D. suzukii*, only the 24-h extract caused a significant decline in survival (*p* = 0.0092), whereas the 48-h extract significantly prolonged survival (*p* < 0.0001) and the 96-h extract had no significant effect compared to the MRS extract control (Figure 5B). These data suggest that the injection of *L. pseudomesenteroides* extracts has a negligible effect in drosophilids.

### 3.7. Oral Administration of L. pseudomesenteroides Extracts to A. pisum

The effects of the *L. pseudomesenteroides* extracts on aphids were tested by diluting the concentrates to a 1× working solution in the advanced AP3 medium, with the extract of sterile MRS medium used as a control. The 24-h extracts had no significant effect on survival, whereas the 48-h and 96-h extracts significantly reduced survival (*p* < 0.0001) resulting in a mortality rate of 40% after 3 days (Figure 6A). The extracts therefore appear to show a species-dependent effect that also becomes more potent with the duration of bacterial cultivation.

### 3.8. Septic Administration of L. pseudomesenteroides Extracts to A. pisum

Finally, we tested the injection of *L. pseudomesenteroides* extracts into aphids. The extracts of the bacterial cultures reconstituted in PBS had no significant effect compared to the mock-injection with the similarly reconstituted extract of sterile MRS medium. There was a slight but nonsignificant increase in survival for aphids injected with the 24-h extract and a decrease in survival for aphids injected with the 48-h and 96-h extracts (Figure 6B).

## 4. Discussion

Biological pest control involves the use of natural predators, parasitoids or pathogens to reduce the size of pest populations. This is envisaged as an ecologically sustainable means to protect food and feed crops without harming beneficial insects. Pathogens have been widely used as biological control agents for insects [11,27,28,29], particularly the bacterium *B. thuringiensis* [64,65] and the fungus *B. bassiana* [11]. We previously showed that oral infection with the lactic acid bacterium *L. pseudomesenteroides* was able to kill the invasive fruit pest *D. suzukii* [54]. Here, we tested this pathogen against three target insects (two drosophilids and one aphid) via the oral and septic infection routes, in each case testing the effect of live bacteria at different concentrations and the effect of cell-free extracts representing different culture durations.

In agreement with our earlier study, the oral intake of live bacteria significantly reduced the survival of *D. suzukii* in a dose-dependent manner. The model fruit fly *D. melanogaster* was also susceptible but to a lesser degree, with a significant effect observed only at the highest dose. One potential explanation is that *D. melanogaster* feeds on rotting fruit, which harbors many species of bacteria, whereas *D. suzukii* feeds on ripening fruit and would not encounter a large number of microbial pathogens in the wild. It is therefore likely that *D. melanogaster* has been placed under more selection pressure in its ecological niche and has evolved a more robust immune response to control bacterial infections [66].

Interestingly, we found that *D. melanogaster* lived longer than *D. suzukii* in all control cohorts, even though previous studies indicate that *D. suzukii* lives longer than *D. melanogaster* under laboratory conditions [67], with a lifespan of 50–154 days [68] compared to 20–52 days [31]. The broad range reflects various factors, such as the health and nutritional status of the flies and the season [69]. Our experimental flies were maintained on a soybean/cornmeal medium diet until the experiments but then on 50 mM sucrose during the exposure and 100 mM sucrose for recovery, which may explain the shorter than usual lifespan of both species in our control cohorts. The cellular immune system of both drosophilid species is similar in structure and response [70] and depends in part on the microbiome [71]. The bacterial communities associated with insects vary by diet and habitat even within the same species, conferring immunity against different location-specific groups of pathogens [72,73]. We tested established laboratory strains of both drosophilid species, and previous studies have shown that laboratory strains often harbor more limited bacterial communities than wild counterparts because the standardized conditions and diet exert little selective pressure for microbiome diversity [74,75]. However, the advantage of standardized conditions and a simple diet is their ability to reveal any intrinsic differences in the virulence of *L. pseudomesenteroides* towards these two drosophilid species. The greater virulence of this bacterium towards the invasive pest *D. suzukii* compared to the model *D. melanogaster* is desirable because it allows the development of host-restricted biocontrol agents.

Septic infection by direct thoracic injection also caused a significant impact on the survival of *D. suzukii* but the outcome was not as clear-cut as the oral infection. Unlike the oral infection, we did not observe a dose-dependent effect, suggesting the active principle is effective at low concentrations once it has gained access to the body. Septic infection had an inconclusive effect on *D. melanogaster*. This was surprising, because *D. suzukii* produces five times as many hemocytes as *D. melanogaster* [70] and should therefore repel invading bacteria more effectively. *L. pseudomesenteroides* was previously shown to cross the intestinal epithelium in both species but accumulates in higher numbers in *D. suzukii*, indicating some form of species-dependent virulence [43]. Most hemocytes are located in the hemolymph as a first line of defense against infections after wounding or against parasitoid wasps [76]. The abundance of hemocytes in the hemolymph of drosophilids may explain the weaker effect of septic vs. oral infection. Furthermore, the injection of 4.6 nL delivers far fewer bacteria than the oral route [77]. However, the higher efficacy of oral infection compared to septic infection with *L. pseudomesenteroides* contrasts with data reported for *Serratia marcescens*, where the test insects (*D. melanogaster*) succumbed within 24 h after septic infection but died much more slowly following oral infection [78].

In addition to the two drosophilid species, we also tested the model pest aphid *A. pisum*. Aphids are sap-sucking pests that damage crops by feeding and by transmitting pathogens [1,2,3,4,30]. They are primarily controlled using chemical insecticides [2,79] but widespread use has increased selection pressure, leading to the evolution of multiple resistant forms, thus creating a demand for new control agents [9,79]. Septic infection with *L. pseudomesenteroides* killed all aphids within 3 days, whereas oral infection was less effective but nevertheless reduced overall survival by 25–30% after 3 days. Entomopathogenic fungi have been used successfully against aphid populations but bacterial control agents have received comparatively little attention [80,81]. *Erwinia aphidicola* is known to infect *A. pisum* and reduce its lifespan [82]. Now we can confirm that *L. pseudomesenteroides* also shows efficacy against this pest and could, in the future, be developed as a biocontrol agent. Interestingly, the effect of *L. pseudomesenteroides* against aphids was dose-independent, suggesting that even small quantities of the bacterium would be useful for population control.

The mechanisms used by *L. pseudomesenteroides* to kill the three target species in this study are unclear. Aphids benefit from a close symbiotic relationship with bacteria, 90% of which belong to the genus *Buchnera* [83]. These bacteria are located in specialized aphid cells (bacteriocytes) and play an essential role in growth and reproduction [84]. It is unlikely that *L. pseudomesenteroides* fills a niche in the aphid microbiome and is ignored by the immune system because the granulocytes recognize Gram-negative and Gram-positive bacteria [85] (the latter including *L. pseudomesenteroides* [86]) and destroy them by phagocytosis. Interestingly, *Leuconostoc* spp. produce bacteriocins, such as leucocin and its derivatives, which kill other bacteria [87,88]. Given that both drosophilids and aphids depend on their microbiome for immunity, it is possible that the virulence of *L. pseudomesenteroides* involves the production of such bactericidal metabolites. Lactic acid bacteria also synthesize diacetyl as a metabolic byproduct, and this is known to inhibit of the growth of Gram-negative bacteria and fungi [89]. The microbiome of *D. melanogaster* features various Gram-negative bacteria, including many acetic acid bacteria (*Acetobacter* spp.) [75], and the primary endosymbionts of *A. pisum* (*Buchnera* spp.), [90] as well as important facultative symbionts, such as *Serratia symbiotica* [91], which are also Gram-negative. Diacetyl can enter Gram-negative bacterial cells via membrane porins without altering membrane permeability [92]. The proposed mode of action relies on the interaction of diacetyl with periplasmic proteins that bind arginine, thus interfering with amino acid metabolism [92]. The absence of similar arginine-binding proteins in Gram-positive bacteria may confer greater resistance to this compound [93]. In our future studies, we will investigate whether the virulence of *L. pseudomesenteroides* involves the suppression of essential bacterial endosymbionts by the production of bacteriocins, diacetyl or other molecules.

To determine whether small soluble compounds, such as secondary metabolites, can replicate the effect of live bacteria, we prepared *L. pseudomesenteroides* organic extracts to denature all proteins and small lipids. For the drosophilid feeding assays, we reconstituted the extracts in DMSO before diluting in sucrose because this is an excellent dipolar aprotic solvent with low toxicity, whereas the extracts were directly reconstituted in the AP3 medium for aphid feeding because aphids are extremely sensitive to DMSO. We found that the extracts had negligible effects against drosophilids when administered either orally or by injection, suggesting that the *L. pseudomesenteroides* virulence mechanism in these species is dependent on the activity of living cells, or on macromolecular components such as protein toxins that are denatured during extraction. Further studies will therefore be needed to identify the active principle in living cells, their site of action in the host and the response pathways that are affected. The secretion of proteins by bacteria is affected by factors such as cell density, stress and nutrients [94,95,96] and it would be interesting to test the virulence of *L. pseudomesenteroides* under diverse conditions (temperature, cell density and media with different nutrient compositions). In contrast, we found that the extracts were effective against aphids in feeding assays (at least the 48-h and 96-h extracts) but none of the extracts showed any significant effect via septic injury, suggesting the presence of a soluble component active against Hemiptera but not Diptera. The higher toxicity of the older cultures suggests that the active principle accumulates in the broth during exponential growth or is strongly induced as the cells enter their stationary phase. As Kolter argues, “Stationary-phase cultures hold unexplored terrain awaiting those interested in almost any area of bacterial physiology” [97] and one potential explanation is that bacterial stress-response pathways triggered by nutrient depletion lead to the accumulation of metabolites that are toxic toward aphids. It is unclear why oral administration is effective but septic injection is not, given the direct damage caused by injection, and further studies are necessary to investigate this phenomenon in more detail.

Taken together, our results show that *L. pseudomesenteroides* was generally more effective when administered as living cells to all three species, although the cell-free extracts showed some limited oral efficacy against aphids. Interestingly, the most effective administration route for living cells was species-dependent, with the oral route more effective in drosophilids and the septic route more effective in aphids. In drosophilids, *L. pseudomesenteroides* therefore behaves as a classic entomopathogen that might be encountered in the wild and consumed along with a fruit meal. The observed efficacy reflects the impact of *L. pseudomesenteroides* along the oral pathway and gut, but the insecticidal effects diminish once the bacterium crosses the gut epithelium and reaches the hemolymph. The active principle was not preserved in the organic extracts, suggesting that the primary effect of the bacterium is based on one or more protein components that may act directly against the insect host or suppress the growth of endosymbionts. In aphids, the live bacteria were more effective when delivered via the septic route, possibly because the cellular immune response of aphids is weaker than the intestinal immune response due to the need to support an obligate endosymbiont population. The mode of food intake differs greatly between aphids and flies, with flies taking food from surfaces that are likely to be teeming with microbes, whereas aphids use piercing mouthparts to penetrate plant tissues and withdraw phloem sap. In contrast, the cell-free extracts had no significant effect when injected but the 48-h and 96-h extracts were effective when delivered orally. This indicates that *L. pseudomesenteroides* metabolites either directly or indirectly induce an insecticidal mechanism in aphids but not in drosophilids. These results highlight the importance of testing the insecticidal activity of entomopathogens in different insect species to better understand their mode of action.

## 5. Conclusions

We have shown that *L. pseudomesenteroides*, a lactic acid bacterium isolated from *D. suzukii*, has significant potential as a novel biocontrol agent for insect pests. We found that *L. pseudomesenteroides* live cells reduced the survival of two drosophilid and one aphid species, although the most effective administration route was species-dependent. Furthermore, *L. pseudomesenteroides* cell-free extracts were able to suppress the growth of aphids when delivered orally. These results provide insight into the efficacy and potential target pests of new biocontrol agents based on the bacterium *L. pseudomesenteroides*.

## Figures and Tables

**Figure 1 insects-11-00471-f001:**
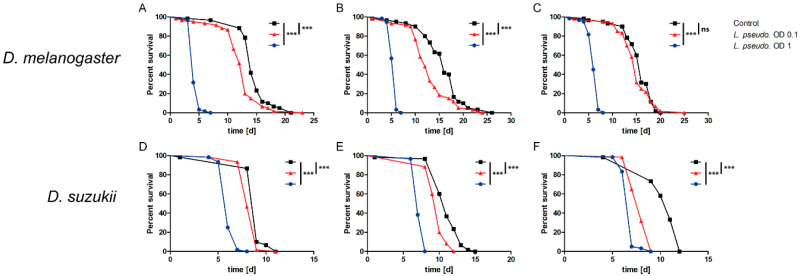
Survival curves for the oral infection of *Drosophila melanogaster* and *D. suzukii* with *Leuconostoc pseudomesenteroides.* Adult flies (3–10 days old) were fed on bacterial suspensions with an optical density (OD_600_) of 1 or 0.1, or a sucrose control. Each survival assay was carried out three times (biological replicates) with three technical replicates (n = 540). A logrank test was used for statistical analysis. (**A**–**C**) All three biological replicates of survival assays with *D. melanogaster*. (**D**–**F**) All three biological replicates of survival assays with *D. suzukii*. Statistical significance is indicated as follows: *p* ≤ 0.0009 ***; ns, not significant.

**Figure 2 insects-11-00471-f002:**
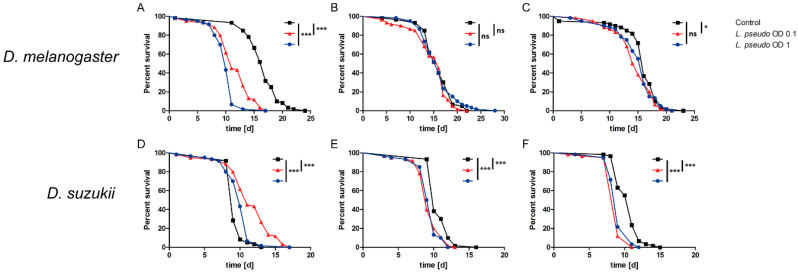
Survival curves for the septic infection of *Drosophila melanogaster* and *D. suzukii* with *Leuconostoc pseudomesenteroides*. Adult flies (3–10 days old) were injected with 4.6 nL of *L. pseudomesenteroides* suspension (OD_600_ = 1 or 0.1) or a PBS control. Each survival assay was carried out three times (biological replicates) with three technical replicates (n = 540). A logrank test was used for statistical analysis. (**A**–**C**) All three biological replicates of survival assays with *D. melanogaster*. (**D**–**F**) All three biological replicates of survival assays with *D. suzukii*. Statistical significance is indicated as follows: *p* ≤ 0.0009 ***; *p* ≤ 0.09 *; ns, not significant.

**Figure 3 insects-11-00471-f003:**
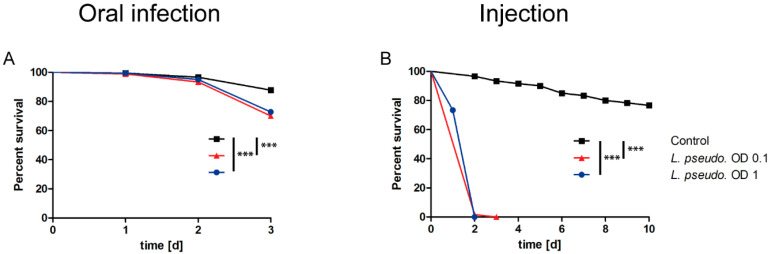
Survival curves for the oral and septic infection of *Acyrthosiphon pisum* nymphs (5 days old) with *Leuconostoc pseudomesenteroides*. (**A**) Oral infection with *L. pseudomesenteroides* suspension (OD_600_ = 1 or 0.1) or sterile advanced AP3 medium as a control followed by survival analysis for 3 days. We tested 540 aphids in total (n = 180 per treatment). (**B**) Injection of nymphs with 25 nL of *L. pseudomesenteroides* suspension (OD_600_ = 1 or 0.1) or PBS as a control followed by survival analysis for 10 days. For the injection, each treatment represents a total of 60 aphids (n = 180). Survival data were tested for significance using the logrank test. Each graph is a composition of three biological replicates. Statistical significance is indicated as follows: *p* ≤ 0.0009 ***.

**Figure 4 insects-11-00471-f004:**
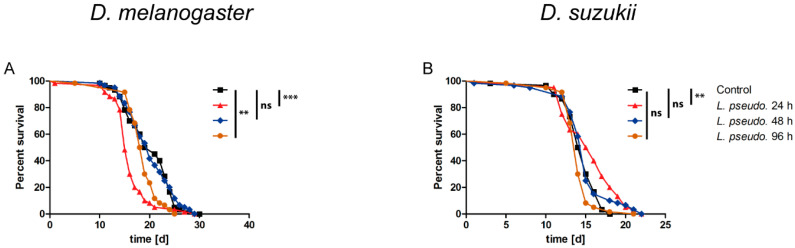
Survival curves for the oral administration of *Leuconostoc pseudomesenteroides* extracts to (**A**) *Drosophila melanogaster* and (**B**) *D. suzukii*. Adult flies (3–10 days old) were starved for 5–6 h before *L. pseudomesenteroides* extracts (12 droplets, each 3 µL) were presented for 2–3 h. The extracts were prepared after bacterial cultivation for 24, 48 or 96 h, and an extract of sterile culture medium was used as a control. A logrank test was used for statistical analysis. Each graph represents three biological replicates, each comprising three technical replicates (n = 180 flies per treatment, making a total of 720 flies per species). Statistical significance is indicated as follows: *p* ≤ 0.0009 ***; *p* ≤ 0.009 **; ns, not significant.

**Figure 5 insects-11-00471-f005:**
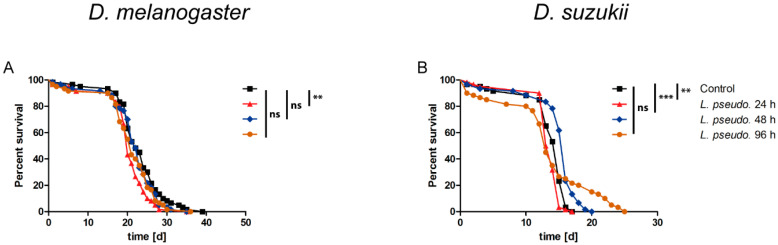
Survival curves for the septic administration of *Leuconostoc pseudomesenteroides* extracts by the injection of (**A**) *Drosophila melanogaster* and (**B**) *D. suzukii* with 4.6 nL of extract prepared after 24, 48 or 96 h, or an extract of sterile culture medium as a control. A logrank test was used for statistical analysis. Each composite graph represents n = 180 flies per treatment, making a total of 720 flies per species. Statistical significance is indicated as follows: *p* ≤ 0.0009 ***; *p* ≤ 0.009 **; ns, not significant.

**Figure 6 insects-11-00471-f006:**
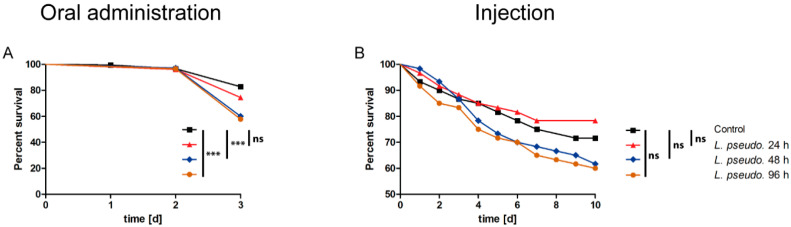
Survival curves for the oral and septic administration of *Leuconostoc pseudomesenteroides* extracts to *Acyrthosiphon pisum* nymphs (5 days old). (**A**) Oral administration of extracts prepared after bacterial cultivation for 24, 48 or 96 h, or an extract of sterile culture medium as a control, reconstituted in the advanced AP3 medium. Each treatment group represents n = 180 aphids (720 in total). (**B**) Injection of the same extracts reconstituted in PBS. Each treatment group represents n = 60 aphids (240 in total). Survival data were tested for significance using the logrank test. Each graph represents a composition of three biological replicates. Statistical significance is indicated as follows: *p* ≤ 0.0009 ***; ns, not significant.

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
