# Peer review of "The Gram-Positive Bacterium Leuconostoc pseudomesenteroides Shows Insecticidal Activity against Drosophilid and Aphid Pests"

_insects, 2020, doi:10.3390/insects11080471_

Round 1
Reviewer 1 Report
This manuscript describes novel possibility for biocontrol of drosophilid and aphid pests using bacterium L. pseudomesenteroides. Authors bring new results about its efficiency against D. suzukii and A. pisum as representative pest for each taxa and use also D. melanogaster as model organism. Methods include several approaches of pest infection (oral versus injection, live or killed bacteria, and bacterial extracts) and results to survival assays.
In general this research is focused on practical approach to observe new biocontrol agent and thus the survival or mortality of hosts is the most important parameter. In this point of view are the concentrations used realistic for field applications? However I would welcome also more details about its mode of action – which immune mechanisms are targeted, which pathways? D. melanogaster is an ideal model for that. Also several other questions should be answered: Authors use only one real pest specie from each group; it is enough to extrapolate this efficiency also to other species? Why only female flies were used, do the males react differently? Females were not mated, is there the difference between mated or not-mated female? Why the experiments were carried out on adults and not on the larvae? What is the idea of bio-control in the field - to spray live/dead bacteria or their extracts on the crop and the pests will get it orally? Aphids are sap-sacking pests, how they can get the sufficient dose to be controlled?
The style of the manuscript is adequate and I consider this manuscript after answering my question as suitable for publication in Insects.
Comments (I can’t use the line numbers, because they are missing):
- Drosophilid x drosophilid, please make it uniform
- 2.1. in fly food is present probably nipagin, not nipalgin
- 2.3. What is the origin of L. pseudomesenteroides, it is some collection strain with accession number available also for other researches?
- Graphs: Why the graphs show three biological replicates separately or one graph is a composition of three biological replicates? This should be explained or uniform.
- Discussion: It is written “The mechanism used by L. pseudomesenteroides to kill A. pisum is unclear” but I think it is the same also for D. suzukii or D. melanogaster, there is no exact mechanism described.
Author Response
Comments and Suggestions for Authors
This manuscript describes novel possibility for biocontrol of drosophilid and aphid pests using bacterium L. pseudomesenteroides. Authors bring new results about its efficiency against D. suzukii and A. pisum as representative pest for each taxa and use also D. melanogaster as model organism. Methods include several approaches of pest infection (oral versus injection, live or killed bacteria, and bacterial extracts) and results to survival assays.
In general this research is focused on practical approach to observe new biocontrol agent and thus the survival or mortality of hosts is the most important parameter. In this point of view are the concentrations used realistic for field applications?
We confirm that the concentrations used in our study are comparable to feed applications previously reported for Bacillus thuringiensis (Pedersen et al., 1994; Canad J Microbiol; doi 10.1139/m95-016). In our study, OD600 = 1 corresponds to 4.9 × 108 CFU/ml, whereas a concentration of 6.5 × 107 CFU/ml B. thuringiensis var. kurstaki (~10 fold lower) was used in the earlier study.
However I would welcome also more details about its mode of action – which immune mechanisms are targeted, which pathways? D. melanogaster is an ideal model for that.
We addressed these points in our previous publication (Hiebert et al., 2020; J Invertebr Pathol; doi 10.1016/j.jip.2020.107389), which is cited in the current manuscript. Leuconostoc pseudomesenteroides blocks food intake by Drosophila melanogaster and D. suzukii larvae, as reported for Pseudomonas entomophila (Vodovar et al., 2005; Proc Natl Acad Sci USA; doi 10.1073/pnas.0502240102), and crosses the intestinal barrier to spread in the hemolymph in the absence of a cellular response. For D. suzukii, our previous study included qPCR experiments, which indicated the induction of drosomycin-like genes encoding putative antimicrobial peptides. As a Gram-positive bacterium, we anticipate that L. pseudomesenteroides triggers a response via the Toll pathway, with drosomycin-like peptides as effectors.
We agree that D. melanogaster would be an ideal model for further studies addressing the mode of action and host response specifically against this Leuconostoc strain.
Also several other questions should be answered: Authors use only one real pest specie from each group; it is enough to extrapolate this efficiency also to other species?
We conducted a side-by-side analysis of three different insect species, which offers insight into the differences between two species within one genus as well as insects from different orders. We do not think it is possible to extrapolate our results to insect pests from not-tested orders and we have been careful not to do so.
Why only female flies were used, do the males react differently?
We use female flies because they are more robust than males. Therefore, any treatments that are effective against females are likely to be more effective against males, which die off more rapidly probably due to their lower energy reserves.
Females were not mated, is there the difference between mated or not-mated female?
There is no difference between mated versus non-mated females.
Why the experiments were carried out on adults and not on the larvae?
The experiments were carried out on adults because these are the targets of the proposed control agents. D. suzukii larvae are well protected within the ripening fruit and are therefore difficult to reach. Our intention is to target adults by spraying or trapping.
What is the idea of bio-control in the field - to spray live/dead bacteria or their extracts on the crop and the pests will get it orally? Aphids are sap-sacking pests, how they can get the sufficient dose to be controlled?
For B. thuringiensis, GFP-marked living bacteria have been shown to enter the roots and migrate through the xylem to all parts of the plant, so that insect pests feeding on other plant tissues are affected (Monnerat et al., 2009; Microbiol Biotechnol; doi 10.1111/j.1751-7915.2009.00116.x). We argue that extracts and even live bacteria could therefore be transported from the roots to parts of the plant that are accessed by aphids.
The style of the manuscript is adequate and I consider this manuscript after answering my question as suitable for publication in Insects.
Comments (I can’t use the line numbers, because they are missing):
- Drosophilid x drosophilid, please make it uniform
Lower case is used throughout other than in the title and section titles (which capitalize all major words by default as per the journal style) and at the beginning of sentences.
- 2.1. in fly food is present probably nipagin, not nipalgin
We thank the reviewer for highlighting this error and we have corrected it.
- 2.3. What is the origin of L. pseudomesenteroides, it is some collection strain with accession number available also for other researches?
We have added the following sentence for clarity “The L. pseudomesenteroides strain was isolated from moribund D. suzukii larvae as previously described (Hiebert et al., 2020).” This is an in-house Fraunhofer collection strain and is available to other researchers upon request.
- Graphs: Why the graphs show three biological replicates separately or one graph is a composition of three biological replicates? This should be explained or uniform.
The two different graphical outputs had following reasons. We started with single biological replicates at the beginning of the study with the oral and injection assays in flies, where we knew from previous work that live fed bacteria are showing some range of differences in the survival kinetics that we wanted to discuss. We decided for clarity reasons to switch to a composite display to reduce the number of graphs for those assays were we had similar survival kinetics.
Discussion: It is written “The mechanism used by L. pseudomesenteroides to kill A. pisum is unclear” but I think it is the same also for D. suzukii or D. melanogaster, there is no exact mechanism described.
We agree with the reviewer and have amended the description to state that the mechanisms used to kill all three target species are unclear.
Reviewer 2 Report
See comments within the manuscript.

Author Response
The manuscript deals with a recently isolated strain of Leuconostoc pseudomesenteroides whose insecticidal activity has previously been documented. The present work assays the same bacterium on additional species and this appears to be the actual original aspect.
Major concerns:
- The introduction needs to be revised with more updated information. Many aspects, especially on the available results of studies with insect pathogenic bacteria should be considered.
We thank the reviewer for this recommendation and have added a section summarizing more recent research involving insect pathogenic bacteria.
- Materials and methods lack important information on the bacterial fractions used, which needs to be clarified.
We have addressed this point below, in response to comments about the materials an methods section.
- No analyses were conducted on the bacterial fractions used in bioassays. No cell counts. No biochemical analysis. So that no conclusions can be reached on the mode of action. All this discussion part should be revised accordingly.
The cell counts were recorded (OD600 = 1 corresponds to 4.9 × 108 CFU/ml) so we have added the following sentence to section 2.4 for clarity: “L. pseudomsesenteroides cell counts revealed that OD600 = 1 corresponded to 4.9 × 108 CFU/ml.”. We formulated our discussion carefully to speculate on the mode of action, for example “…suggesting the presence of a soluble component active against Hemiptera but not Diptera” and “…The active principle was not preserved in the organic extracts, suggesting that the primary effect of the bacterium is based on one or more protein components that may act directly against the insect host or suppress the growth of endosymbionts”. The italicized words were used deliberately to make it clear that these are conditional rather than definitive statements.
- Discussion should be deeply revised according to the actual finding of this work and not speculating on the mode of action that was not investigated appropriately. In addition, this is a preliminary study in the laboratory and no information are provided on the actual potential in semi-field or field condition, or even on the safety of the bacterium so that no conclusion can be reached on the potential of it in biological control.
We disagree with this comment. It is acceptable for the discussion section of a research article to speculate on the broader implications of the results as long as it is made clear that the statements are speculative, and the language conventions we used in the manuscript serve this purpose (see previous comment). We agree that this study does not provide a ready-to-use solution and that the data are preliminary, but we acknowledge this throughout the paper (especially in the discussion) and accordingly we do not consider the discussion in need of extensive revision.
Other comments.
The manuscript is missing line numbers.
INTRODUCTION
- Not all chemical insecticides are so hazardous, provided they are used correctly. The are part of the IPM principles. These statements should be carefully revised.
We agree and have modified the sentence as follows: “…although some chemical insecticides are not hazardous when applied correctly,…”
- Include in the introduction more updated information on insect pathogenic bacteria used in integrated pest management, and state of the art research (updated information should be included).
We have added a section providing more recent information about insect pathogenic bacteria as requested.
- The author says that no information are available on the effect of bacterial organic acids on insects. And speculate this is one of the original aspects of their study. This not true. For instance, see the following paper demonstrating the insecticidal activity of organic acids produced by Lactobacillus sp.
Torres, M. J., Rocha, V. F., & Audisio, M. C. (2016). Laboratory evaluation of L actobacillus johnsonii CRL 1647 metabolites for biological control of M usca domestica. Entomologia Experimentalis et Applicata, 159(3), 347-353.
See also the following paper on the metabolite produced by foodborne lactic acid bacteria on insect pathogenic bacteria that may help on the discussion on the wide bioactive compound diversity these kinds of bacteria produce:
Lazzeri, A. M., Mangia, N. P., Mura, M. E., Floris, I., Satta, A., & Ruiu, L. (2020). Potential of novel food-borne Lactobacillus isolates against the honeybee pathogen Paenibacillus larvae. Biocontrol Science and Technology, 1-12.
We thank the reviewer for this information and have revised the manuscript as follows: “Although lactic acid bacteria are known to interfere with the growth of phytopathogenic bacteria and fungi [48,49] and also entomopathogenic bacteria (Lazzeri et al., 2020), little is known about their potential as biocontrol agents against insect pests (Torres et al., 2016).”
MATERIALS AND METHODS:
- It is not clear how bioassays were conducted with aphids. How were they feed, considering their sucking mouth parts.
This is already addressed in section 2.6. The aphids were presented with parafilm and were “…able to pierce the Parafilm and suck up the bacterial suspensions or extracts.”
- How was the cell free fraction obtained? How were cells eliminated? By centrifugation? This is not specified in the description. Did you check if no living cells were still present? If so how? This is important because you have bioassays in which no differences in effects are concentration dependent, so that even a low number of cells remaining in the “cell-free” fraction might be the cause of the observed effect.
We have revised the manuscript to provide more information. The cells were eliminated by incubation in methanol for 2 h, followed by centrifugation and reconstitution in 100% DMSO. To check that no living cells were present, small aliquots of the reconstituted extracts were incubated overnight on MRS plates, and no colonies were formed. The revised text reads as follows: “For the preparation of cell-free extracts, 0.5-ml aliquots of the pre-culture were transferred to 300-ml Erlenmeyer flasks containing 50 ml MRS medium as above, and incubated for 24, 48 or 96 h. In each case, the broth was lyophilized and extracted with methanol. The lyophilized culture was incubated for 2 h with methanol while shaking. Afterwards, cell debris was separated by centrifugation and extract supernatants were transferred to a fresh vessel for drying under reduced pressure. To confirm the absence of cells, an aliquot of the extract reconstituted in 100% DMSO was incubated on MRS agar plates. No colonies grew after overnight incubation at 28°C.”
The 16S rDNA sequence of the bacterium that is mentioned should be deposited in a public repository (eg. NCBI GenBank) and the accession number provided in the manuscript.
We deposited the 16S rDNA sequence as requested (accession number MT724349, release pending, scheduled July 12th 2020). We have added this information to section 2.8.
- Ways of administration of bacterial preparations to insects needs to be clarified in respect to the following result presentation. See below.
RESULTS AND DISCUSSION:
- Be clear and consistent with the definitions of the bacterial fractions used in bioassays. You have a cell fraction and a fell free supernatant. Why to talk about and extract? What is it about? I haven’t found a description of an extraction process in materials and methods.
We have amended the description of the extraction process in section 2.3 and the term “extract” as used throughout the manuscript refers to this.
- Regarding the way of bacterial administration to insects I suggest to talk about: injection, contact and ingestion. What do you mean with septic administration?
The term “administration” is used for the extract because it has no bacterial cells and is therefore not infectious. Septic administration is the injection extract and oral administration is the mixing of extract with food. In contrast, septic infection is the injection of living cells and oral infection is the mixing of living cells with food.
- I haven’t seen in this work any analysis to determine a possible septicaemia (bacterial replication in the host). Death of the host may depend just on the activity of metabolites and toxins, but this has not been demonstrated with specific experiments or it appears to be so.
We do not mention or intent to analyze septicemia in this manuscript. The term “septic infection” is used solely to indicate the infection route (direct injection) to contrast with oral infection (via food) with similar terminology for the extracts (see previous point). This nomenclature is consistent with our previous study (Hiebert et al., 2020; J Invertebr Pathol; doi 10.1016/j.jip.2020.107389).
- First line of discussion: the definition of BCA should be revised including predators, parasitoids, and all microbial agents.
We have included parasitoids along with natural predators and pathogens in the first sentence of the discussion as requested.
- Statistical output should be reported for data in each of the figures.
We have added the statistical analysis to the figures as requested.
FIGURES
The figure legend should be revised stating that they represent “Survival curves….”
We have added this information as requested
Reviewer 3 Report
The manuscript deals with a recently isolated strain of Leuconostoc pseudomesenteroides whose insecticidal activity has previously been documented. The present work assays the same bacterium on additional species and this appears to be the actual original aspect.
Major concerns:
- The introduction needs to be revised with more updated information. Many aspects, especially on the available results of studies with insect pathogenic bacteria should be considered.
- Materials and methods lack important information on the bacterial fractions used, which needs to be clarified.
- No analyses were conducted on the bacterial fractions used in bioassays. No cell counts. No biochemical analysis. So that no conclusions can be reached on the mode of action. All this discussion part should be revised accordingly.
- Discussion should be deeply revised according to the actual finding of this work and not speculating on the mode of action that was not investigated appropriately. In addition, this is a preliminary study in the laboratory and no information are provided on the actual potential in semi-field or field condition, or even on the safety of the bacterium so that no conclusion can be reached on the potential of it in biological control.
Other comments.
The manuscript is missing line numbers.
INTRODUCTION
- Not all chemical insecticides are so hazardous, provided they are used correctly. The are part of the IPM principles. These statements should be carefully revised.
- Include in the introduction more updated information on insect pathogenic bacteria used in integrated pest management, and state of the art research (updated information should be included).
- The author says that no information are available on the effect of bacterial organic acids on insects. And speculate this is one of the original aspects of their study. This not true. For instance, see the following paper demonstrating the insecticidal activity of organic acids produced by Lactobacillus sp.
Torres, M. J., Rocha, V. F., & Audisio, M. C. (2016). Laboratory evaluation of L actobacillus johnsonii CRL 1647 metabolites for biological control of M usca domestica. Entomologia Experimentalis et Applicata, 159(3), 347-353.
See also the following paper on the metabolite produced by foodborne lactic acid bacteria on insect pathogenic bacteria that may help on the discussion on the wide bioactive compound diversity these kinds of bacteria produce:
Lazzeri, A. M., Mangia, N. P., Mura, M. E., Floris, I., Satta, A., & Ruiu, L. (2020). Potential of novel food-borne Lactobacillus isolates against the honeybee pathogen Paenibacillus larvae. Biocontrol Science and Technology, 1-12.
MATERIALS AND METHODS:
- It is not clear how bioassays were conducted with aphids. How were they feed, considering their sucking mouth parts.
- How was the cell free fraction obtained? How were cells eliminated? By centrifugation? This is not specified in the description. Did you check if no living cells were still present? If so how? This is important because you have bioassays in which no differences in effects are concentration dependent, so that even a low number of cells remaining in the “cell-free” fraction might be the cause of the observed effect.
- The 16S rDNA sequence of the bacterium that is mentioned should be deposited in a public repository (eg. NCBI GenBank) and the accession number provided in the manuscript.
- Ways of administration of bacterial preparations to insects needs to be clarified in respect to the following result presentation. See below.
RESULTS AND DISCUSSION:
- Be clear and consistent with the definitions of the bacterial fractions used in bioassays. You have a cell fraction and a fell free supernatant. Why to talk about and extract? What is it about? I haven’t found a description of an extraction process in materials and methods.
- Regarding the way of bacterial administration to insects I suggest to talk about: injection, contact and ingestion. What do you mean with septic administration? I haven’t seen in this work any analysis to determine a possible septicaemia (bacterial replication in the host). Death of the host may depend just on the activity of metabolites and toxins, but this has not been demonstrated with specific experiments or it appears to be so.
- First line of discussion: the definition of BCA should be revised including predators, parasitoids, and all microbial agents.
- Statistical output should be reported for data in each of the figures.
FIGURES
The figure legend should be revised stating that they represent “Survival curves….”
Round 2
Reviewer 3 Report
The manuscript has been improved